# Space-time risk cluster of visceral leishmaniasis in Brazilian endemic region with high social vulnerability: An ecological time series study

Caique J. N. Ribeiro[1], Allan D. dos Santos[2], Shirley V. M. A. Lima[1,2], Eliete R. da Silva[3], Bianca V. S. Ribeiro[4], Andrezza M. Duque[1], Marcus V. S. Peixoto[5], Priscila L. dos Santos[1,4], Iris M. de Oliveira[6], Michael W. Lipscomb[7], Karina C. G. M. de Araújo[1,4], Tatiana R. de Moura[1,4] *

1 Health Sciences Graduate Program, Federal University of Sergipe, Aracaju, Brazil, 2 Department of Nursing, Federal University of Sergipe, Lagarto, Brazil, 3 Department of Nursing, Federal University of Sergipe, Aracaju, Brazil, 4 Graduate Program in Parasite Biology, Federal University of Sergipe, São Cristóvão, Brazil, 5 Department of Speech Therapy and Audiology, Federal University of Sergipe, São Cristóvão, Brazil, 6 Department of Functional Biology and Health Sciences, University of Vigo, Pontevedra, Spain, 7 Department of Biology, Howard University, Washington DC, United States of America

* tmoura.ufs@gmail.com

**Data Availability Statement:** All relevant data are within the manuscript and its Supporting information files.

## Abstract

### Background

Despite visceral leishmaniasis (VL) being epidemic in most Brazilian regions, the Northeast region is responsible for the highest morbidity and mortality outcomes within the country.

### Objective

To analyse the spatiotemporal dynamics of VL cases to identify the temporal trends and high-risk areas for VL transmission, as well as the association of the disease with social vulnerability in Brazilian Northeast.

### Methods

We carried out an ecological time series study employing spatial analysis techniques using all VL confirmed cases of 1,794 municipalities of Brazilian Northeast between the years 2000 to 2017. The Social Vulnerability Index (SVI) was used to represent the social vulnerability. Incidence rates were standardized and smoothed by the Local Empirical Bayesian Method. Time trends were examined through segmented linear regression. Spatiotemporal analysis consisted of uni- and bivariate Global and Local Moran indexes and space-time scan statistics.

### Results

Incidence rate remained stable and ranged from 4.84 to 3.52 cases/100,000 inhabitants. There was higher case prevalence between males (62.71%), children and adolescents (63.27%), non-white (69.75%) and urban residents (62.58%). Increasing trends of new cases

**Funding:** This work was supported by National Council for Scientific and Technological Development of Brazil (CNPq) (MCTI/CNPQ/28/2018: 434623/2018-0) www.cnpq.br (TRdeM); Support Foundation for Research and Technological Innovation of the State of Sergipe (FAPITEC) (MS/CNPq/FAPITEC/SE/SES/850226/2017: 019.203.00933/2018-0) www.fapitec.se.gov.br/ (TRdeM); Coordination for the Improvement of Higher Education Personnel (CAPES) through Graduate Support Program of Federal University of Sergipe (PROAP/UFS) (Processes: N° 23113.062217/2019-50, 23113.062290/2019-21, and 23113.062214/2019-16) www.gov.br/capes/ (TRdeM, PLdosS, KCGMdeA); and U.S. National Institutes of Health (SC1GM127207) https://www.nih.gov/ (MWL). The funders had no role in study design, data collection and analysis, decision to publish, or preparation of the manuscript.

**Competing interests:** The authors have declared that no competing interests exist.

were observed among adult male subjects ($\geq$ 40 years old) and urban residents. Importantly, VL incidence showed a direct spatial dependence. Spatial and space-time clusters were identified in *sertão* and *meio-norte* sub-regions, overlapping with high social vulnerability areas.

## Conclusions

VL is a persistent health issue in Brazilian Northeast and associated with social vulnerability. Space-time clustering of VL cases in socially vulnerable municipalities demands intersectoral public policies of surveillance and control, with focus on reducing inequalities and improving living conditions for regional inhabitants.

### Author summary

Visceral leishmaniasis (VL) remains a worldwide health issue, with increasing rates of mortality being observed. Brazil has an epidemiological scenario of expanding VL transmission, especially in the Northeast region. In the present study, we analysed spatiotemporal dynamics of VL cases and its association with social vulnerability in Brazilian Northeast. Briefly, data was analysed of all VL confirmed cases during the years of 2000 to 2017 and the Social Vulnerability Index (SVI) from 1,794 municipalities of Brazilian Northeast. Results revealed that VL continues to spread heterogeneously, with space-time high-risk clusters in the most socially vulnerable areas. We observed increasing trends of new cases among male subjects $\geq$ 40 years of age and urban residents. Our study represents the first investigation that demonstrates associations between VL and social vulnerability in the Northeast region of Brazil. These findings could contribute to VL prevention, surveillance, and control through better understanding of disease distribution, affording effective prioritization of municipalities with higher vulnerability. Thus, reduction of social inequality and better living conditions should be part of the planning of public health policies related to VL control.

## Introduction

Despite the epidemiologic transition in Brazil, infectious diseases, especially neglected tropical diseases (NTDs), are still a major public health problem [1], such as visceral leishmaniasis (VL) that can be life-threatening if not properly treated [2].

It is estimated that one billion people live in VL-endemic regions worldwide, with 300,000 new cases and 20,000 deaths per year due to the disease. However, 94% of the diagnosed cases are concentrated in only six countries; among them, Brazil [3] was responsible for 97% of the recorded incidents in South America in 2017 [4]. In this continent, *Leishmania infantum* protozoan is the disease-causing agent transmitted by bites from female phlebotomies, and domestic dogs are the main urban reservoir [5].

The epidemiology of VL has changed dynamically due to a myriad of interactions among environmental, socioeconomic, demographic and immunological factors [6]. In Brazil, VL was previously considered a rural endemic disease. However, in recent decades, because of urbanization, the majority of cases have occurred in large cities and the surrounding urban areas [7].

In the 1990s, approximately 90% of recorded VL cases occurred in the Northeast region of Brazil. By the 2000s, the disease spread to other Brazilian urban regions [8]. Furthermore, a recent study demonstrated that 56% of VL deaths between 2000 to 2011 occurred within the Northeast region [9].

VL incidence has been connected to social inequality and poor living conditions [10–13]. Brazil is considered one of the most unequal countries in terms of wealth distribution. Further compounding the situation, the Northeast has important socioeconomic disparities, which are represented by the highest social vulnerability index (SVI) and the lowest human development index (HDI) in the country [14]. This is in addition to being endemic for several NTDs [15]. The SVI shows relative access, absence or insufficiency of services, which include some basic needs that should be ensured for all citizens [16].

Therefore, investigating VL cases associated with the social vulnerability factors could support appropriate health interventions for specific regional conditions. Thus, this research aimed to analyse the spatiotemporal dynamics of VL cases to identify the temporal trends and high-risk areas for VL transmission, as well as the association of the disease with social vulnerability in Brazilian Northeast.

## Methods

### Ethics statement

This study used public-domain aggregate secondary data and followed national and international ethical recommendations, as well as the rules of the Helsinki Convention. All data analysed were anonymized. The research project was approved by the Research Ethics Committee of Federal University of Sergipe (CEP/UFS), registered under the approval number 2,537,671.

### Study design

This is an ecological time series study that used spatial analysis techniques including all confirmed VL cases in the Northeast region of Brazil between 2000 and 2017. Units of the analysis were the 1,794 municipalities in the region.

### Study area description

The Northeast region of Brazil (latitude: 01˚02'30" N/18˚ 20' 07" S; longitude: 34˚47'30" /48˚ 45'24" O) is divided into four subregions (*meio-norte*, *sertão*, *agreste* and *zona da mata*). This corresponds to 18% of the national territory (Fig 1) with an estimated population of 57 million inhabitants [17].

### Data sources

Morbidity data were collected from *Sistema de Informação de Agravos de Notificação* (SINAN) of the *Departamento de Informática do Sistema Único de Saúde* (Datasus) [18]. The Brazilian Northeastern population estimates and cartographic base (shapefile extension), presented in the latitude/longitudinal system (SIRGAS 2000), were obtained from *Instituto Brasileiro de Geografia e Estatística* (IBGE) [17]. SVI was taken from *Instituto de Pesquisa Econômica Aplicada* (IPEA) database (www.ipea.gov.br). The data used in this study is available in S1 Data.

### Variables and measures

The primary measurement of study was the VL incidence rate in municipality level. This rate was obtained by dividing the average of cases by the central population of each municipality and multiplying by 100,000. We used VL transmission risk stratification of the Brazilian Guide to Health Surveillance (2019) as follows: sporadic transmission ($<$2.4 cases/100,000 inhabitants), moderate transmission ($\geq$2.4 and $<$4.4 cases/100,000 inhabitants) and intense

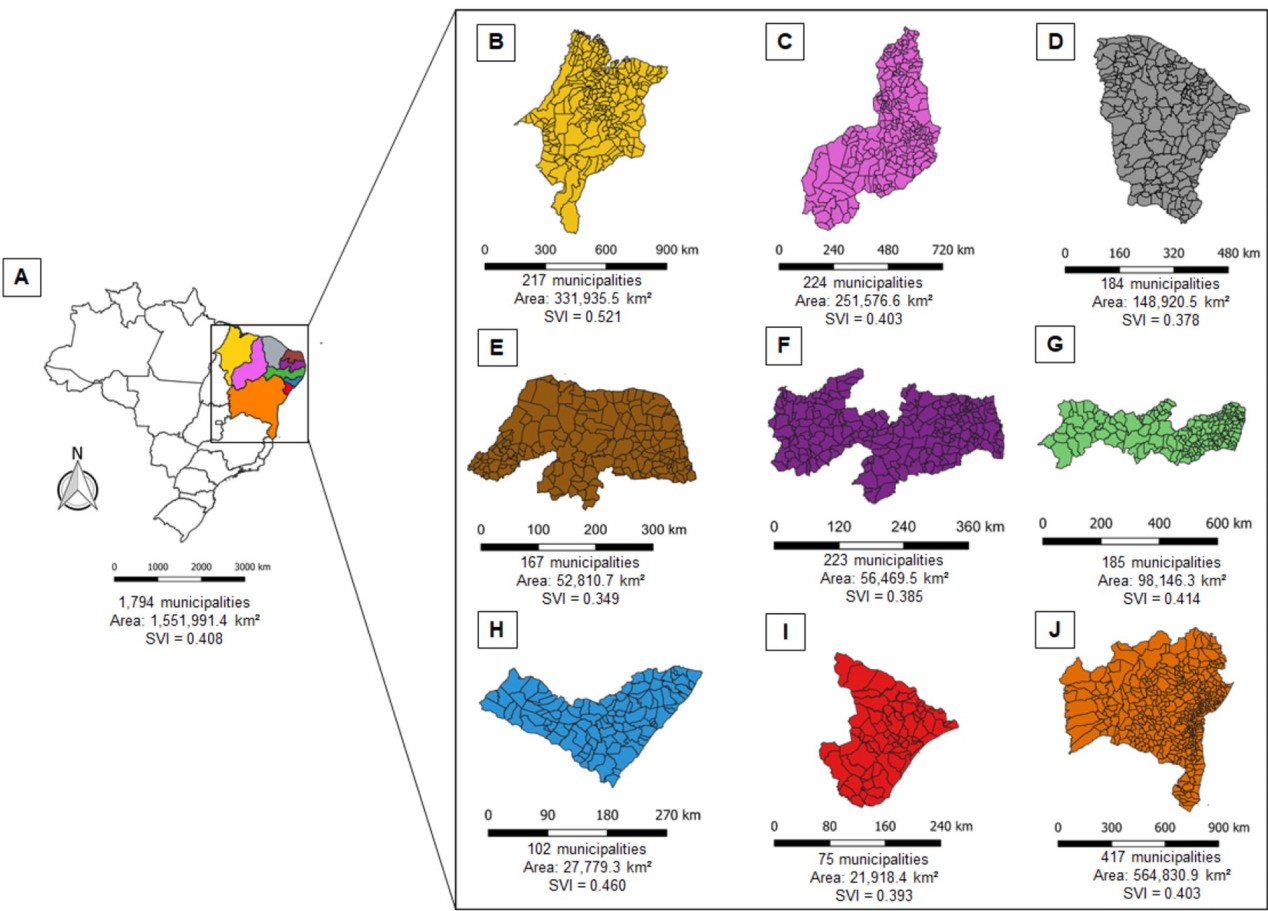

**Fig 1. Study area.** (A) Northeast–NE, Brazil. (B) Maranhão–MA. (C) Piauí–PI. (D) Ceará–CE. (E) Rio Grande do Norte–RN. (F) Paraíba–PB. (G) Pernambuco–PE. (H) Alagoas–AL. (I) Sergipe–SE. (J) Bahia–BA. SVI: Social vulnerability index.

transmission ($\geq$4.4 cases/100,000 inhabitants) [5]. VL incidence rates was also calculated in state and regional levels. The classification and description of all variables used in our analysis are shown in Table 1.

SVI was employed as an independent variable for the occurrence of VL transmission in the municipalities of Brazilian Northeast. This index estimates exclusion and vulnerability beyond insufficient monetary resources and is composed of 16 indicators from data of Census 2010 grouped into three dimensions (urban infrastructure–SVI-UI; human capital–SVI-HC; and income/work–SVI-I/W) (S1 Table). Each dimension has the same weight for calculating the global SVI. Furthermore, all indicators were normalized into a scale varying from 0 to 1, in which 0 corresponds to an ideal situation and 1, to the worst situation. The complete construction methodology is described in the official report from IPEA [14].

The SVI-UI dimension reflects the conditions that affect quality of life, such as access basic sanitation services and urban mobility. Health conditions and access to education were used to determine individual prospects and are composed of the SVI-HC dimension. Insufficient family income, adult unemployment, informal employment of poorly educated adults, family dependence on elderly income and child labour indicate household income security, which is presented through the SVI-I/W dimension [14]. The SVI ranges from 0 to 1; the closer the

**Table 1. Description of study variables.**

| Classification | | Variable | Description | Analysis |
|---|---|---|---|---|
| Dependent | Primary | Incidence rate (per 100,000 inhabitants) | Calculated in municipality, state, and regional levels | Time trends Spatial cluster Spatiotemporal cluster Bivariate spatial cluster |
| | Secondary | Prevalence rate (per 100,000 inhabitants) | Calculated in state and regional levels | Time trends |
| | | Mortality rate (per 100,000 inhabitants) | Calculated in regional level | |
| | | Lethality | Calculated in regional level | |
| Independent | | Social vulnerability index (SVI) | Very low (0 to 0.200), low (0.201 to 0.300), medium (0.301 to 0.400), high (0.401 to 0.500), and very high ($\geq$ 0.501) | Bivariate spatial cluster |
| | | Year of occurrence | 2000 to 2017 | Time trends |
| Explanatory | | State of residence | Alagoas, Bahia, Ceará, Maranhão, Paraíba, Pernambuco, Piauí, Rio Grande do Norte, and Sergipe | Descriptive epidemiological characterization |
| | | Sex | Male and female | |
| | | Age group | $\leq$ 4 years, 5–19 years, 20–39 years, 40–59 years, and $\geq$ 60 years | |
| | | Ethnicity/skin colour | White and non-white | |
| | | Residence zone | Rural, urban and periurban | |
| | | VL-HIV co-infection | | |
| | | Level of education | < 8 years and $\geq$ 8 years | |
| | | Case type | New case, relapse, and transference | |
| | | Clinical outcome | Cure, abandonment, death, and transference | |

index is to 1, the greater the social vulnerability of a municipality, which is classified as very low (0 to 0.200), low (0.201 to 0.300), medium (0.301 to 0.400), high (0.401 to 0.500) and very high ($\geq$ 0.501) [16].

## Time trends analysis

Time trends were examined by segmented linear regression (Joinpoint), based on the calculation of the annual percentage changes (APCs), calculated for each segment, and average annual percentage changes (AAPCs) for the entire period when there was more than one significant inflexion in a study period, with their respective 95% confidence interval (95%CI). Monte Carlo permutation test was used to obtain the statistical significance, applying 999 permutations, and choose the best number of significant segments. APCs and AAPCs were significative when p<0.05 and their 95%CIs did not include zero. The selected final model was the most adjusted, allowing the best representation of trend, with the fewest number of inflexion points [19]. The results were interpreted as follows: positive and significant APCs/AAPCs were considered increasing trends, negative and significant APCs/AAPCs were considered decreasing trends; on the other hand, when there was no significance, the trend was considered stable [20,21].

## Spatial cluster analysis

First, crude VL incidence rates were smoothed by applying the local Bayesian empirical method to correct the random fluctuations and provide more stability to the incidence rates [22]. Crude and smoothed rates were represented on maps stratified by the risk of VL transmission [5].

The global Moran's I index was computed to analyse the spatial autocorrelation using a first order proximity matrix, which was expanded upon using contiguity criterion. This index ranges from -1 to +1, with positive values indicating positive spatial autocorrelation and

negative values indicating negative autocorrelation. Additionally, values close to zero point the lack of spatial autocorrelation [23]. Statistical significances were identified using Monte Carlo simulations with 999 permutations.

Once the autocorrelation was identified, the local Moran's index (LISA) was used to indicate the occurrence of spatial clusters of municipalities with high VL transmission [24] and to generate a scatter plot with four quadrants: Q1 (municipalities with high VL incidence rates and high incidence rates in neighbouring municipalities), Q2 (municipalities with low VL incidence rate and low incidence rate in neighbouring municipalities), Q3 (municipalities with high VL incidence rates and low incidence rates in neighbouring municipalities) and Q4 (municipalities with low VL incidence rates and high incidence rates in neighbouring municipalities). The diagram was depicted through Moran maps, in which only the statistically significant results were considered (p<0.05).

## Spatiotemporal cluster analysis

Kulldorff's retrospective space-time scan statistical analysis was performed to identify high-risk spatiotemporal clusters for VL transmission and to estimate the relative risks (RRs) of VL occurrence for each cluster in relation to its neighbours [25]. The Poisson's discrete probability model was used for scanning since the single events under analysis (VL cases) are counts and considered rare [21], under the null hypothesis that the expected number of cases in each area is proportional to its population size [26]. We established the following conditions for the model, according to previous studies on infectious diseases [27,28]: aggregation time of 1 year, no geographical overlap of clusters, circular clusters, maximum spatial cluster size of 50% of the at-risk population, and a maximum temporal cluster of 50% of the study period. The primary (or most likely) and secondary clusters were detected using the log likelihood ratio (LLR) test and were represented through choropleth maps. The results were statistically significant when p<0.05 using 999 Monte Carlo simulations.

## Bivariate spatial cluster analysis

Initially, we represented the spatial distribution of the social vulnerability in Northeast region of Brazil through choropleth maps for SVI and its dimensions (SVI-IU, SVI-HC, and SVI-I/W). Subsequently, we performed the univariate global Moran's I index to analyse the spatial autocorrelation of social vulnerability, using a first order proximity matrix. Univariate LISA analysis was applied to identify clustering and the significant results (p<0.05) were depicted in Moran maps, which were visually compared with the results of spatial and spatiotemporal cluster analysis of VL incidence rates [24].

In order to verify the association between the occurrence of VL and social vulnerability, Spearman's correlation test was performed to examine correlation(s) between the VL incidence rate and SVI and its dimensions (SVI-IU, SVI-HC, and SVI-I/W). As there was positive correlation between VL incidence rate and SVI, we investigated the existence of spatial correlation between VL transmission and social vulnerability using a bivariate analysis of global Moran's index and LISA.

Similar to univariate analysis, the bivariate global Moran's does not reveal spatial clustering [13,29]. Thus, the bivariate LISA analysis was employed to determine the degree of spatial correlation of the data in relation to its neighbours [23], generating a scatter plot with four quadrants [30]: Q1 (municipalities with high VL incidence rates and high social vulnerability in neighbouring municipalities), Q2 (municipalities with low VL incidence rate and low social vulnerability in neighbouring municipalities), Q3 (municipalities with high VL incidence rates and low social vulnerability in neighbouring municipalities) and Q4 (municipalities with low

VL incidence rates and high social vulnerability in neighbouring municipalities). These clusters were depicted in Moran maps and only the statistically significant results were considered (p<0.05).

## Softwares

Microsoft Office Excel 2016 (Microsoft Corporation; Redmond, WA, EUA) was used to store and prepare the data. QGis 3.4.11 (QGIS Development Team; Open Source Geospatial Foundation Project) was used to produce choropleth maps. TerraView 4.2.2 (www.inpe.br) and GeoDa 1.14 [30] were employed to perform spatial analysis. Joint Point Regression 4.6 (US National Cancer Institute, Bethesda, MD, EUA) were used to time trend analysis. SaTScan 9.6 (Harvard Medical School, Boston and Information Management Service Inc., Silver Spring, MD, EUA) was used to analyse spatiotemporal clusters.

## Results

A total of 36,514 VL cases were confirmed in Brazilian Northeast between 2000 and 2017. Fig 2 describes the number of cases per federative unit/state and the respective annual frequencies and average prevalence rates. Maranhão state had a high number of registered cases in this period, corresponding to 28.86% of the total records.

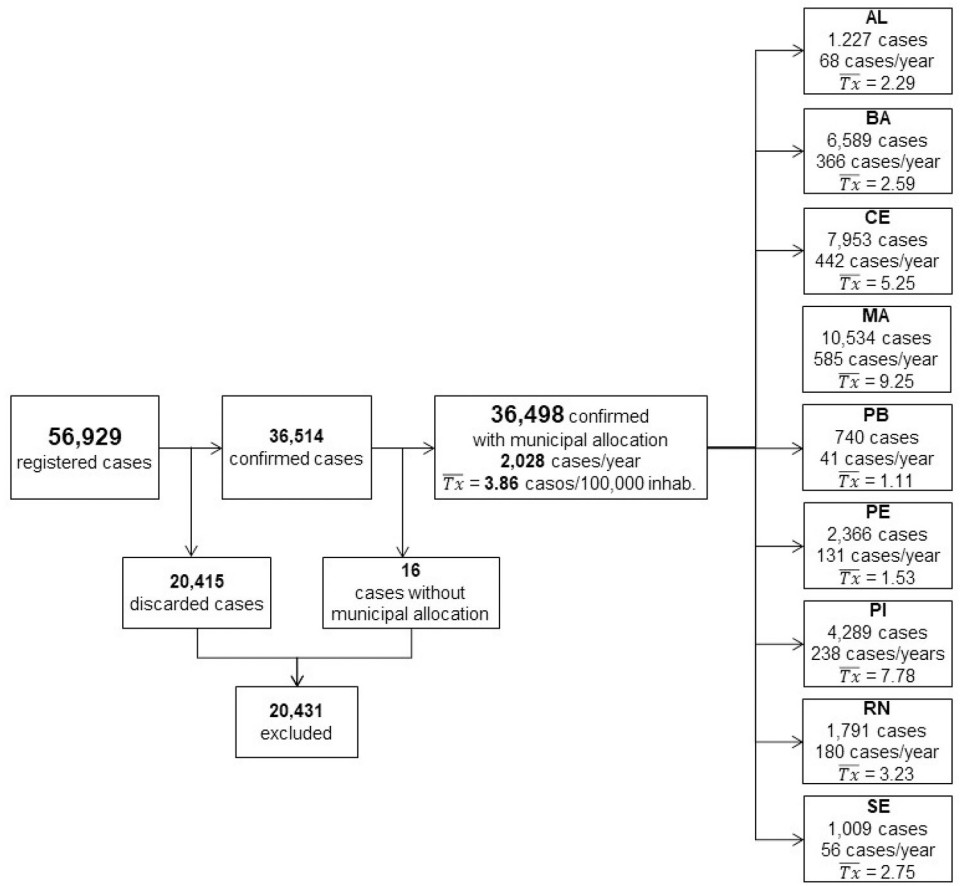

**Fig 2. Flowchart of study population describing the number of cases, average frequencies, and annual prevalence rates per state.** $\overline{Tx}$: average prevalence (cases/100,000 inhabitants/year).

Table 2 shows the baseline characteristics of the VL epidemiologic indicators. The predominant characteristics of the VL cases in Brazilian Northeast were males (62.71%), <5 years old (40.77%), non-white (69.75%), urban residents (62.58%) with a low education level (27.01%) and residents who were cured (71.15%). Only in the federative unit/state of Alagoas were there more prevalent cases in rural areas (69.19%). The epidemiological characterization per state is available in S2 Table.

### Time trends analysis

Until 2010, the proportion of municipalities with VL transmission remained stable, but since then, an annual increase of 3.6 (95%CI: 1.0 to 6.4; p<0.05) was observed (S1 Fig). Table 3 highlights the inflexion points for trend changes of VL epidemiological indicators. The crude prevalence rate in the general population ranged from 5.57 in 2000 to 3.36 cases per 100,000 inhabitants in 2017, with an annual decrease of -1.4 (95%CI: -2.6 to -0.2; p<0.05), while the annual incidence remained stable, varying from 4.84 in 2000 to 3.52 cases per 100,000 inhabitants in 2017 (p>0,05). In turn, Alagoas and Rio Grande do Norte showed decreasing trends in the numbers of new cases, whereas there was an increasing trend in Ceará (Table 2). The incidence rates showed increasing trends in males (APC: 1.4; 95%CI: 0.8 to 2.0; p<0.05) and groups of individuals between ages of 40–50 years old (APC: 3.8; 95%CI: 2.2 to 5.4) and ≥ 60 years old (APC: 5.9; 95%CI: 4.2 to 7.6 p<0.05). Furthermore, there were statistically significant increases in the percentages of cases with VL-HIV co-infections and in the crude mortality rate (AAPC: 2.5; 95%CI: 0.6 to 4.4; p<0.05) and lethality (APC: 3.9; 95%CI: 3.0 to 4.9; p<0.05) (Table 3).

### Spatial cluster analysis

VL transmission was broadly distributed, as shown in Fig 3A. Almost a quarter of municipalities (429) had intense transmission (≥ 4.4 cases per 100,000 inhabitants). When the smoothed rates were considered through the local empirical Bayesian method (Fig 3B), results revealed that this amount was up to 30% in the study area (542 municipalities). New cases were concentrated mainly in the *sertão* and *meio-norte* sub-regions, which comprise the states of Bahia, Maranhão, Pernambuco and Piauí. The global Moran's I index showed a significant spatial autocorrelation (0.338, p = 0.001), highlighting spatial dependence of new VL cases in municipalities with similar patterns. Fig 3C presents the municipalities identified through the LISA analysis. The high-risk clusters were detected in 269 municipalities of Maranhão (96), Piauí (67), Bahia (60), Ceará (29), Alagoas (12) and Pernambuco (5).

### Spatiotemporal cluster analysis

The space-time scan statistics identified 12 significant spatiotemporal clusters of new VL cases in the general population (p<0.001), as shown in Table 4 and illustrated in Fig 3D. The primary cluster included 465 municipalities and the highest number of cases (8,245), from 2000 to 2008, in the federative units/states of Piauí (200), Maranhão (119), Bahia (97), Ceará (28) and Pernambuco (21), with a crude incidence rate of 9.5 per 100,000 inhabitants (RR = 3.35; p<0.001). Importantly, Bahia's municipalities contained 7 of 12 identified clusters, with one municipality from Bahiahaving the highest annual incidence rate (81.6 cases/100,000 inhabitants) and the highest relative risk (RR = 23.61; p<0.001).

### Bivariate spatial analysis

Fig 4A shows the distribution of social vulnerability in Brazilian Northeast through SVI (SVI-UI, SVI-HC, and SVI-I/W). Approximately 76% of municipalities had high to very high

**Table 2. Baseline characteristics.**

| Variables | n = 36,498 | % |
|---|---|---|
| **Federative unit / state** | | |
| Alagoas | 1,227 | 3.36 |
| Bahia | 6,589 | 18.05 |
| Ceará | 7,953 | 21.79 |
| Maranhão | 10,534 | 28.86 |
| Paraíba | 740 | 2.03 |
| Pernambuco | 2,366 | 6.48 |
| Piauí | 4,289 | 11.75 |
| Rio Grande do Norte | 1,791 | 4.91 |
| Sergipe | 1,009 | 2.76 |
| **Case type** | | |
| New cases | 32,862 | 90.03 |
| Relapse | 1,404 | 3.85 |
| Transference | 447 | 1.22 |
| Miss data | 1,785 | 4.90 |
| **Sex** | | |
| Male | 22,888 | 62.71 |
| Female | 13,587 | 37.23 |
| Miss data | 23 | 0.06 |
| **Age** | | |
| 0–4 years | 14,880 | 40.77 |
| 5–19 years | 8,211 | 22.5 |
| 20–39 years | 7,539 | 20.65 |
| 40–59 years | 4,252 | 11.65 |
| $\geq$ 60 years | 1,599 | 4.38 |
| Miss data | 17 | 0.05 |
| **Ethinicity / skin colour** | | |
| White | 3,446 | 9.44 |
| Nonwhite | 25,458 | 69.75 |
| Miss data | 7,594 | 20.81 |
| **Zone** | | |
| Urban | 22,840 | 62.58 |
| Rural | 12,190 | 33.4 |
| Periurban | 400 | 1.09 |
| Miss data | 1,068 | 2.93 |
| **Level of education** | | |
| < 8 years | 9,859 | 27.01 |
| $\geq$ 8 years | 3,066 | 8.4 |
| Miss data/N.A. | 23,573 | 64.59 |
| **Outcome** | | |
| Cure | 25,970 | 71.15 |
| Abandonment | 144 | 0.39 |
| Death | 2,690 | 7.37 |
| Transference | 1,849 | 5.07 |
| Miss data | 5,845 | 16.01 |
| N.A. not applicable | | |

**Table 3. Time trends of VL epidemiologic indicators.**

| Indicator/variable | Segmented period | | | Entire period | |
|---|---|---|---|---|---|
| | Period | APC (95%CI) | Trend | AAPC (95%CI) | Trend |
| Crude prevalence rate (per 100,000 inhab.) | | | | | |
| | 2000–2017 | -1.4 (-2.6 to 0.2) | Decreasing | | |
| Crude incidence rate (per 100,000 inhab.) | | | | | |
| General | 2000–2017 | -1.22 (-2.5 to 0.1) | Stable | | |
| Federative Unit/State | | | | | |
| Alagoas | 2000–2007 | -26.2 (-33.5 to -18.1) | Decreasing | -10.6 (-16.3 to -4.6) | Decreasing |
| | 2007–2017 | 2.2 (-7.5 to 12.9) | Stable | | |
| Bahia | 2000–2017 | -1.1 (-3.6 to 1.4) | Stable | | |
| Ceará | 2000–2006 | 24.8 (8.5 to 43.6) | Increasing | 5.5 (0.2 to 10.9) | Increasing |
| | 2006–2017 | -3.8 (-7.6 to 0.2) | Stable | | |
| Maranhão | 2000–2009 | -8.2 (-13.1 to -3.0) | Decreasing | -1.8 (-5.5 to 2.1) | Stable |
| | 2009–2017 | 6.0 (-0.8 to 13.3) | Stable | | |
| Paraíba | 2000–2017 | -2.2 (-6.2 to 2.0) | Stable | | |
| Pernambuco | 2000–2003 | -43.7 (-60.9 to -18.7) | Decreasing | -6.2 (-12.3 to 0.2) | Stable |
| | 2003–2017 | 4.6 (0.3 to 9.1) | Increasing | | |
| Piauí | 2000–2004 | 28.5 (5.8 to 56.0) | Increasing | 3.4 (-3.2 to 10.4) | Stable |
| | 2004–2009 | -13.2 (-26.9 to 3.1) | Stable | | |
| | 2009–2017 | 3.4 (-3.3 to 10.5) | Stable | | |
| Rio Grande do Norte | 2000–2003 | -42.8 (-50.8 to -33.4) | Decreasing | -7.8 (-11.3 to -4.2) | Decreasing |
| | 2003–2011 | 7.4 (1.5 to 13.5) | Increasing | | |
| | 2011–2017 | -4.5 (-10.2 to 1.6) | Stable | | |
| Sergipe | 2000–2002 | -31.6 (-68.6 to 48.9) | Stable | -1.7 (-9.9 to 7.4) | Stable |
| | 2002–2017 | 3.2 (-0.4 to 7.0) | Stable | | |
| Sex | | | | | |
| Male | 2000–2017 | 1.4 (0.8 to 2.0) | Increasing | | |
| Female | 2000–2017 | -2.6 (-3.9 to -1.3) | Decreasing | | |
| Age | | | | | |
| ≤ 4 years | 2000–2017 | -1.4 (-3.0 to 0.3) | Stable | | |
| 5–19 years | 2000–2002 | -23.1 (-41.6 to 1.3) | Stable | -3.6 (-6.6 to -0.5) | Decreasing |
| | 2002–2017 | -0.7 (-2.2 to 0.8) | Stable | | |
| 20–39 years | 2000–2017 | 0.6 (-0.6 to 1.8) | Stable | | |
| 40–59 years | 2000–2017 | 3.8 (2.2 to 5.4) | Increasing | | |
| ≥ 60 years | 2000–2017 | 5.9 (4.2 to 7.6) | Increasing | | |
| Percentage of LV-HIV co-infection | | | | | |
| | 2000–2011 | 24.98 (15.8 to 34.9) | Increasing | 17.3 (11.3 to 23.7) | Increasing |
| | 2011–2017 | 4.5 (-4.2 to 13.9) | Stable | | |
| Crude mortality rate (per 100,000 inhab.) | | | | | |
| | 2000–2009 | -0.9 (-3.8 to 2.0) | Stable | 2.5 (0.6 to 4.4) | Increasing |
| | 2009–2017 | 6.5 (3.4 to 9.7) | Increasing | | |
| Lethality | | | | | |
| | 2000–2017 | 3.9 (3.0 to 4.9) | Increasing | | |
| Proportion of municipalities with transmission | | | | | |
| | 2000–2010 | -0.8 (-2.4 to 0.8) | Stable | 1 (-0.3 to 2.3) | Stable |
| | 2010–2017 | 3.6 (1.0 to 6.4) | Increasing | | |

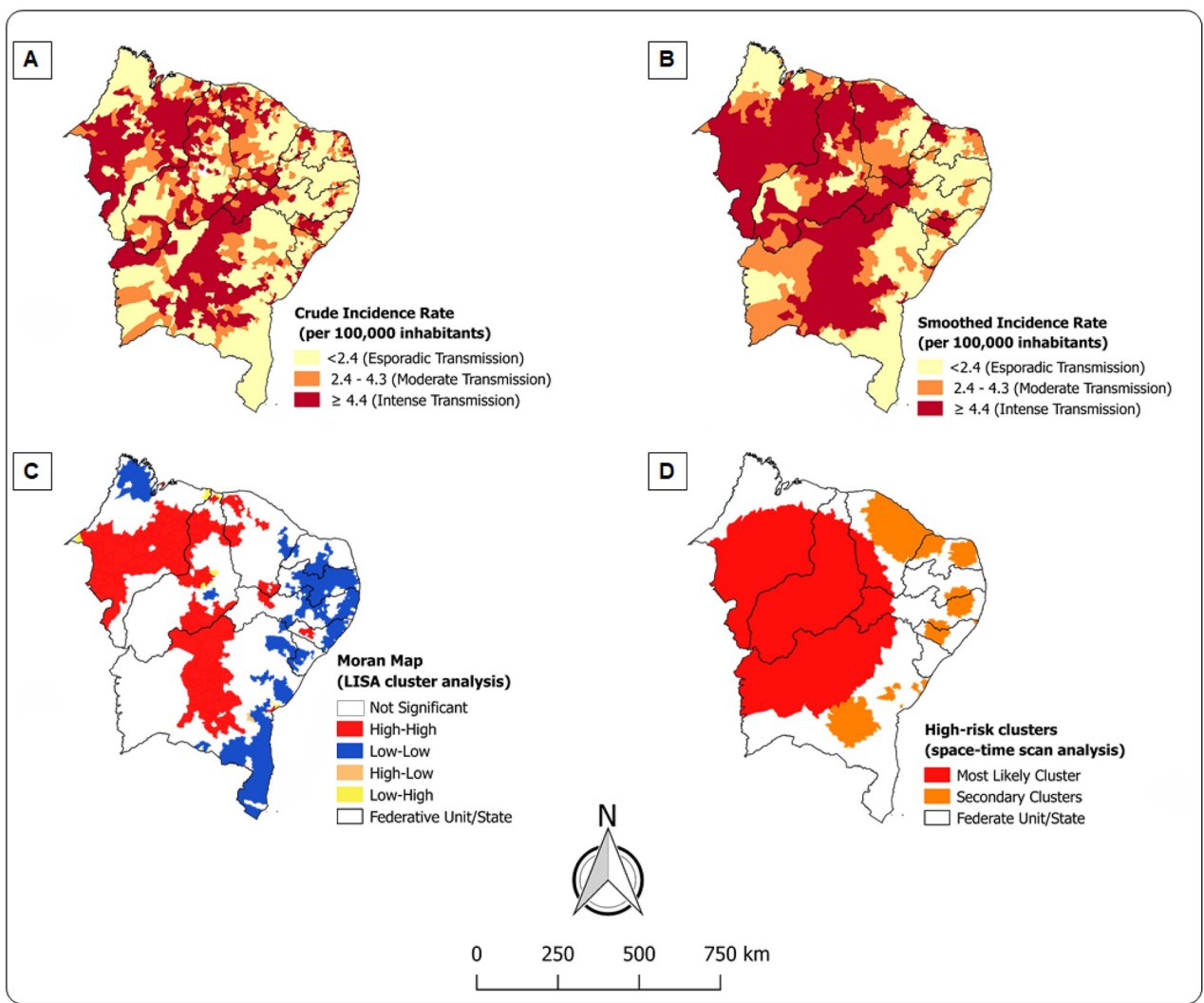

**Fig 3. Spatial and spatiotemporal distribution of VL, Northeast, Brazil (2000–2017). The maps show high-risk clusters for VL transmission mainly in *sertão* and *meio-norte* sub-regions.** (A) VL crude incidence rates. (B) VL smoothed incidence rates. (C) Univariate LISA cluster analysis. (D) Space-time scan statistical analysis.

social vulnerability (858 and 502, respectively), and a quarter (340) had intense VL transmission (Table 5).

Table 6 demonstrates that the global Moran's I index revealed significant spatial autocorrelation in the socially vulnerable municipalities (Fig 4B). A cluster of high social vulnerability (Q1) for all SVI domains was formed in Maranhão. Social vulnerability clusters related to human capital were evidenced in 7 of 9 states in the region. In Bahia, there was a clustering of municipalities with social vulnerability related to income-work aspects.

There was a positive correlation between VL incidence rate and all SVI domains (Table 6). The bivariate analyses of the global Moran's I index and LISA were significant for all aspects of social vulnerability (Fig 4C). In the SVI-UI, SVI-HC and SVI-I/W dimensions, 114, 119 and 107 municipalities, respectively, were VL high-risk clusters. A significant positive correlation was also observed in the general SVI with the clustering of 119 municipalities in Maranhão (Fig 4D). Smaller high-risk clusters were observed in Alagoas, Bahia, Pernambuco and Piauí.

**Table 4. Space-time clusters of annual crude incidence rate of LV per 100,000 general population.**

| Cluster | Time period | Number of municipalities | States | Number of new cases | Expected number of new cases | Annual incidence rate[a] | RR | LLR |
|---|---|---|---|---|---|---|---|---|
| 1 | 2000–2008 | 465 | Maranhão, Piauí, Ceará, Pernambuco, Bahia | 8,245 | 2,991 | 9.5 | 3.35 | 3598.54 |
| 2 | 2006–2014 | 108 | Ceará, Rio Grande do Norte | 3,210 | 1,957 | 5.7 | 1.71 | 361.24 |
| 3 | 2000–2002 | 32 | Pernambuco, Alagoas | 373 | 63 | 20.4 | 5.94 | 353.06 |
| 4 | 2003–2011 | 1 | Bahia | 94 | 4 | 81.6 | 23.61 | 207.07 |
| 5 | 2000–2001 | 62 | Paraíba, Pernambuco | 355 | 105 | 11.7 | 3.42 | 184.20 |
| 6 | 2004–2006 | 33 | Bahia | 272 | 72 | 13.1 | 3.80 | 162.16 |
| 7 | 2000–2001 | 48 | Rio Grande do Norte | 191 | 43 | 15.3 | 4.44 | 136.30 |
| 8 | 2009–2013 | 7 | Bahia | 88 | 25 | 12.3 | 3.55 | 48.28 |
| 9 | 2000–2000 | 2 | Pernambuco, Alagoas | 22 | 1 | 61.1 | 17.66 | 42.41 |
| 10 | 2000–2000 | 1 | Bahia | 20 | 1 | 53.0 | 15.31 | 35.87 |
| 11 | 2002–2007 | 1 | Bahia | 31 | 5 | 22.8 | 6.60 | 32.18 |
| 12 | 2000–2000 | 3 | Bahia | 15 | 1 | 44.0 | 12.72 | 24.32 |

RR: relative risk for the cluster compared with the rest of the region; LLR: likelihood ratio.

[a] LV incidence rate (per 100,000 inhabitants) during the clustering time.

## Discussion

Despite the international, national, and local efforts in recent decades to control and eliminate VL, this disease remains a worldwide health problem. To the best of our knowledge, this is the first study that describes the VL transmission dynamics in Brazilian Northeast and its association with social vulnerability using techniques of spatiotemporal clustering analyses. VL had a heterogeneous geographical distribution with clusters strongly associated with social vulnerability. Several studies have investigated the VL trends and/or their spatial distributions in different states and municipalities in the Northeast region of Brazil, but few have clarified the association between VL and social vulnerability [31–35]. Considering this gap, we carried out an integrative approach, including time trends, spatiotemporal, univariate, and bivariate spatial cluster analysis, in order to strengthen the methodology.

The spatial distributions of infectious diseases in the Brazilian Northeast have been reported, such as those for tuberculosis [36,37], leprosy [28], schistosomiasis [38], zika, dengue [39] and chikungunya [40]. This scenario shows that NTDs are a major health issue in the region and that their characterization is a difficult challenge for public health management. VL incidence remained stable during the 18 years analysed in this study (i.e. 2000–2017). Nevertheless, in the last decade there was a territorial expansion of the disease coupled with an increase in the proportion of municipalities with reported cases.

Along these lines, the states of Maranhão and Ceará are highlighted, with the latter having an increasing trend in the number of new cases. Some regions of Maranhão compound the Legal Brazilian Amazon, which is a strategic place for agrobusiness interests and has been suffering profound anthropic modifications in the ecosystem [41]. Similarly, recent study carried out in Ceará pointed out that the highest VL incidences occur in Sobral and Cariri, where there are rapid and unplanned urbanizing, intense anthropic action and migration [42]. Thus, we hypothesize that the association between the environmental modifications and the deterioration of living conditions of population from Maranhão and Ceará states can partially explain the VL expansion.

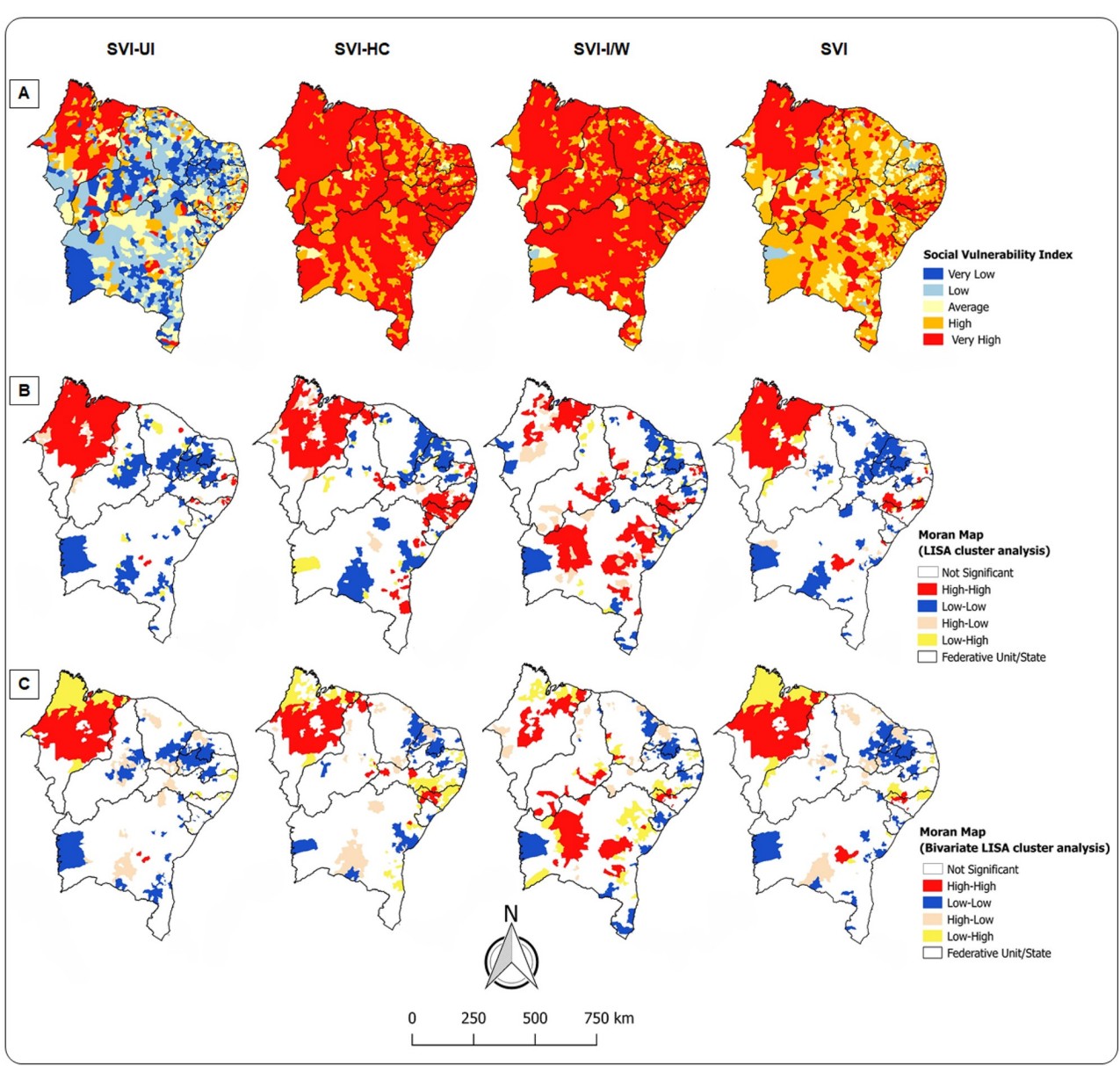

**Fig 4. Association between VL transmission and social vulnerability, Northeast, Brazil (2000–2017).** (A) Spatial distribution of SVI. (B) Univariate LISA cluster analysis of SVI. (C) Bivariate LISA cluster analysis of VL incidence rate and SVI.

In the general population, mortality and lethality were increased, as reported in a nation-wide analysis of epidemiology, trends and spatial patterns of VL case fatality [9]. A possible explanation could be related to the higher prevalence among children under 5 years old and the increasing trend in the number of cases among older adults, especially the elderly given that VL can lead to more severe consequences in these extreme age groups [43,44]. In 2016, the Global Burden Disease's study (GBD) [45] pointed out that VL caused approximately 21 years of life lost (YLL), which reinforces the need for more attention and intervention to reduce fatal-ities. In addition, after an increase in the number of cases with VL-HIV co-infection, this trend

**Table 5. Distribution of Northeastern municipalities of Brazil according to social vulnerability and VL transmission.**

| SVI | Municipalities with VL transmission n (%) | | | Total |
|---|---|---|---|---|
| | Sporadic | Moderate | Intense | |
| Very low | 1 (0.1) | - | - | 1 (0.1) |
| Low | 33 (1.8) | 3 (0.2) | 9 (0.5) | 45 (2.5) |
| Medium | 268 (14.9) | 40 (2.2) | 80 (4.5) | 388 (21.6) |
| High | 497 (27.7) | 158 (8.8) | 203 (11.3) | 858 (47.8) |
| Very high | 287 (16.0) | 78 (4.3) | 137 (7.6) | 502 (28.0) |
| Total | 1,086 (60.5) | 279 (15.6) | 429 (23.9) | 1,794 (100.0) |

**Table 6. Association between VL transmission and social vulnerability in Northeast region of Brazil.**

| Social vulnerability | Spearman's test | | Global Moran's index | | | |
|---|---|---|---|---|---|---|
| | Rho | p-value | Univariate | p-value | Bivariate | p-value |
| SVI | 0.078 | 0.001 | 0.53 | 0.001 | 0.10 | 0.001 |
| SVI-UI | 0.083 | 0.001 | 0.56 | 0.001 | 0.13 | 0.001 |
| SVI-HC | 0.020 | 0.406 | 0.46 | 0.001 | 0.03 | 0.005 |
| SVI-I/W | 0.081 | 0.001 | 0.35 | 0.001 | 0.04 | 0.001 |

remained stable. The coexistence of both diseases complicates the handling of the case and increases mortality, therapeutic failure and relapse rates [32,43,46–51].

Increasing trends in the numbers of new cases were observed among males and urban residents. The higher male susceptibility to VL has been described in animal and human studies, which hypothesize that men are more frequently exposed to outdoor sandflies and higher levels of testosterone [32,52]. In recent decades, VL urbanization has been consolidating in Brazil because of the rural exodus that started between the 1950s and 1980s and population growth in the suburbs with poor living conditions [7]. The higher VL prevalence observed among non-white and less educated subjects supports the hypothesis that social determinants of health, whether at the population or individual level, can structurally influence the disease patterns [53]. The historical heritage of slavery and ineffective public policies of social inclusion mark the Brazilian territory and may partially explain the regional health disparities [54].

There have been investigations into how and why social iniquities affect population health [55]. Therefore, the association between NTDs and poverty has been strongly established [56], since both are socially stigmatizing conditions that create feedback loops [57]. In Bihar, India, it was shown that households with the worst socioeconomic indicators were more affected by VL than households with better socioeconomic indicators [10,58]. Similarly, in Brazil, a significant association between VL and social vulnerability has been reported. In Araguaína, Tocantins, bivariate LISA analysis revealed high-risk cluster for VL incidence in zone with worst vulnerability indicators [13]. Moreover, social determinants have been related to VL mortality, especially in the North and Northeast regions. Recent ecological study highlighted high mortality rates of VL associated with unplanned urbanization and precariousness of households, where both reservoirs and breeding sites for disease vectors are present [59].

Local studies carried out in the Northeast region of Brazil also demonstrate this alarming reality. In Rio Grande do Norte, there is an association between VL incidence and households with no garbage collection or piped water supply [32]. In Aracaju, Sergipe, it was observed that VL is heterogeneously distributed, with higher concentration of cases in outskirts, where there

are risk factors for transmission of vector borne diseases [33]. Yet, a spatial analysis conducted in São Luís, Maranhão, pointed out high prevalence of infection in canine reservoirs in areas of recent occupation with poor sanitary conditions [60].

Although Brazil is the 9th largest economy worldwide, social inequality is a persistent issue, and the Northeast region of Brazil is the most remarkable example of how the development process did not occur simultaneously throughout the country [61]. In this study, spatiotemporal clustering was observed in *sertão* and *meio-norte*, the Northeastern sub-regions with large scarcities of resources and higher social vulnerability. This result corroborates the findings of a study that showed clustering of leprosy cases in the same region [62], which could overwhelm the health system to handle with surveillance of multiple infectious diseases simultaneously. When compared to the national average, all Northeastern states have higher percentages of people living with no water supply or garbage collection, inadequate sanitary sewage, income inequality according to the Gini index, and worse municipal human development index (HDI-M) values. It is important to emphasize that almost a quarter of the residents in Maranhão live in poor conditions; this state had the strongest correlation between VL incidence and the SVI [63]. In case of Bahia state, its municipalities comprised 7 out of 12 spatiotemporal clusters, beyond the remarkable social vulnerability, the broad dispersion of *Lutzomyia longipalpis* throughout the state can be related to the intense VL transmission [64].

In examining our results and previous studies, it is possible to infer that surveillance and control actions have been failing to reduce VL incidence and lethality in Brazil [1], since an epidemiologic scenario of transmission expansion has been maintained [4,65]. Meanwhile, there is a Brazilian policy called the VL Surveillance and Control Program that conducts vector control strategies, reservoir eradication, early diagnosis and timely treatment of cases [5,8]. Unfortunately, the program lacks scientific evidence to prove its cost effectiveness [66,67]. Therefore, we believe that planning and implementation of public policies for VL control should consider the reduction in social iniquities since most of the Northeastern populations live without basic sanitation nor access to health services. In this sense, spatiotemporal techniques could be useful for monitoring and prioritizing high-risk areas, as demonstrated in a research carried out in Belo Horizonte, Minas Gerais, Brazil. The authors investigated the spatiotemporal distribution of VL incidence in humans (HVL) and prevalence of canine VL (CVL) in order to identify priority areas for control actions through Bayesian empirical method, univariate (HVL) and bivariate (HVL *versus* CVL) LISA analysis, and space-time scan statistics [29].

In 2011, the Pan American Health Organization instituted a VL regional program to optimize the surveillance, prevention, and control of leishmaniasis in the Americas. The action plan for the period of 2017–2022 set goals of lethality reduction throughout the continent and a decrease of 50% in incidence rates of countries with expanding VL transmission [65]. According to our findings, Brazil may not achieve this goal. Combating VL and other NTDs should be of high priority to decision makers because their elimination could contribute to reducing disparities of health and socioeconomic statuses in Brazil, thereby improving overall quality of life for its citizens. Further evaluative studies are necessary to identify the critical points in the Brazilian VL program that have hindered the reduction in VL incidence and lethality in Northeast region. Moreover, there is a need to conduct investigations with spatial regression and prospective spatiotemporal cluster analysis to identify the epidemiological and social determinants related to intense clustering of VL cases in states in the Northeastern sub-regions *sertão* and *meio-norte*.

This study has some limitations. First, the ecological design with the use of secondary data did not enable the establishment of casual links. Along those lines, it was only possible to interpret that there was a significant association between VL and social vulnerability. It is important consider the potential underreporting of cases and missingness of some data (i.e.

level of education, outcome, HIV serology, etc.), possibly due to the weaknesses of the health system in the poorest municipalities of Northeast region. Furthermore, SVI has been elaborated with data from the last *census* performed in 2010 and it is known that Brazilian social scenario has been changing because of the neoliberal policies and current global crisis. So, the poverty and social vulnerability could be more profound than the SVI demonstrates.

Despite of these limitations, our study demonstrates a living VL epidemiologic scenario of 18 years and the impactful use and integration of spatial, temporal, and spatiotemporal analysis for disease surveillance and control, as they allow the prioritization of areas with higher transmission risks and the understanding of the potential association of the disease dynamics with the social phenomena in the territory. Additionally, we recognize that the SVI can be an important tool for future health research in Brazil because it is a robust and multidimensional index developed for the purpose of guiding decision making of public managers and help researchers to better understand the different aspects of social vulnerability in Brazil [14]. Some studies applied SVI or its indicators in spatial analysis of leprosy [28,62,68] and VL mortality [59], but they showed promising insights for using the index in epidemiological research.

In conclusion, the results of this study revealed that VL is a persistent health problem in the Brazilian Northeast and that the disease has a strong correlation with social vulnerability. The spatiotemporal clusters indicated the high priorities areas for disease control and suggest interventions for the prevention of VL dissemination to susceptible municipalities or to those with sporadic transmission. In a country such as Brazil, with so much social inequality in health, NTD eradication can only be possible through intersectoral policies that focus on the reduction in inequality and improvements to the living conditions for its population.

## Supporting information

**S1 Data.**
(XLSX)

**S1 Table. Components of social vulnerability index (SVI).**
(DOCX)

**S2 Table. Baseline characteristics per state.**
(DOCX)

**S1 Fig. Time trend of percentage of municipalities with VL transmission showing the disease expansion, especially in the states of Maranhão and Ceará.**
(TIF)

## Author Contributions

**Conceptualization:** Caique J. N. Ribeiro, Allan D. dos Santos, Shirley V. M. A. Lima, Karina C. G. M. de Araújo, Tatiana R. de Moura.

**Data curation:** Caique J. N. Ribeiro, Bianca V. S. Ribeiro.

**Formal analysis:** Caique J. N. Ribeiro, Allan D. dos Santos.

**Funding acquisition:** Priscila L. dos Santos, Michael W. Lipscomb, Karina C. G. M. de Araújo, Tatiana R. de Moura.

**Investigation:** Caique J. N. Ribeiro, Eliete R. da Silva, Bianca V. S. Ribeiro.

**Methodology:** Caique J. N. Ribeiro, Allan D. dos Santos, Karina C. G. M. de Araújo, Tatiana R. de Moura.

**Project administration:** Caique J. N. Ribeiro.

**Resources:** Caique J. N. Ribeiro, Allan D. dos Santos, Tatiana R. de Moura.

**Software:** Caique J. N. Ribeiro, Allan D. dos Santos.

**Supervision:** Tatiana R. de Moura.

**Validation:** Caique J. N. Ribeiro, Allan D. dos Santos, Shirley V. M. A. Lima, Karina C. G. M. de Araújo, Tatiana R. de Moura.

**Visualization:** Caique J. N. Ribeiro.

**Writing – original draft:** Caique J. N. Ribeiro, Tatiana R. de Moura.

**Writing – review & editing:** Caique J. N. Ribeiro, Allan D. dos Santos, Shirley V. M. A. Lima, Andrezza M. Duque, Marcus V. S. Peixoto, Priscila L. dos Santos, Iris M. de Oliveira, Michael W. Lipscomb, Karina C. G. M. de Araújo, Tatiana R. de Moura.

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
