## [Decision Letter · Decision Letter 0]

8 May 2020

Dear Dr. Rodrigues de Moura,

Thank you very much for submitting your manuscript "Space-time risk cluster of visceral leishmaniasis in Brazilian endemic region with high social vulnerability: an ecological and time series study" for consideration at PLOS Neglected Tropical Diseases. As with all papers reviewed by the journal, your manuscript was reviewed by members of the editorial board and by several independent reviewers. In light of the reviews (below this email), we would like to invite the resubmission of a significantly-revised version that takes into account the reviewers' comments. 

We cannot make any decision about publication until we have seen the revised manuscript and your response to the reviewers' comments. Your revised manuscript is also likely to be sent to reviewers for further evaluation.

Sincerely,

Sitara SR Ajjampur, MD, PhD

Guest Editor

Genevieve Milon

Deputy Editor

Reviewer's Responses to Questions

**Key Review Criteria Required for Acceptance?**

**Methods**

-Are the objectives of the study clearly articulated with a clear testable hypothesis stated?

-Is the study design appropriate to address the stated objectives?

-Is the population clearly described and appropriate for the hypothesis being tested?

-Is the sample size sufficient to ensure adequate power to address the hypothesis being tested?

-Were correct statistical analysis used to support conclusions?

-Are there concerns about ethical or regulatory requirements being met?

Reviewer #1: adequado

Reviewer #2: First of all I would like to say thanks for your contributions on this very sensitive issue visceral leishmaniosis in Brazil. 

How can you calculate and measure the VL incidence rate in municipality level? 

Study Variable classifications and definitions is not clear? (Response variable? Explanatory variables? Please summarize in tabular form.

The methodological part is not clear for instance spatiotemporal cluster analysis try to elaborate on the Poisson probability discrete model, bivariate spatial analysis, try to rewrite explicitly including relevant dynamic spatiotemporal models. Try to put also the detail statistical model in short and precise manner.

Reviewer #3: The authors provide a clear and concise overview of their study methodology, evident in their description of the SVI value range and its interpretation. Any mention of a statistical test should be met with a citation to help with reproducibility. Based on Figure 4C, the clustering does not appear to be circular, so what was the authors justification for each chosen scan statistic parameter?

**Results**

-Does the analysis presented match the analysis plan?

-Are the results clearly and completely presented?

-Are the figures (Tables, Images) of sufficient quality for clarity?

Reviewer #1: adequado

Reviewer #2: The result presented are good as descriptive but try to modify the tables 1 for base line characteristics if it is possible you have to present in readable manner and put in the previous page. 

In trend analysis you said it is stationary? How have you check test of stationarity? If so which test are used? If it is unit root what are you going to do with non-stationary trend /data? Please specify and define the test of Stationarity with its hypothesis testing procedures. 

In the result section you said there was a statistical significance? (How and when we say statistically significant) please interpret your results in relation to p-vale. Again put table 2 in previous page. Also put figure 4 on previous page.

Moreover there are a lot of works are done on this area so try to include all recent and relevant works under discussion part including the dynamic spatiotemporal modeling in VL.

Reviewer #3: Reporting unadjusted statistics in Table 1 is important to report. Also reporting population adjusted statistics (e.g., using the population of Brazil an external reference group) would make comparing VL and other characteristics across the ten federative units/states. The authors include helpful detail to interpret the results, however, these details are best for the methods or discussion. Figure 1 does not seem to add any detail to the paper and can be omitted.

**Conclusions**

-Are the conclusions supported by the data presented?

-Are the limitations of analysis clearly described?

-Do the authors discuss how these data can be helpful to advance our understanding of the topic under study?

-Is public health relevance addressed?

Reviewer #1: adequado

Reviewer #2: What are the limitations or gaps for this study? Please clearly state in precise manner. What are the key limitation of analysis? What is the contribution of this study in the area including public health relevance please update this part under discussion section.

Reviewer #3: Based on Appendix 1, Appendix 2, and Appendix 3 the SVIs are likely correlated. Presently, the study does neither assess this correlation, mention it in the discussion, or cite previous studies that may have done so already. 

What characteristics of Maranhão and Ceará may explain why VL incidence is increasing?

The number of study limitations discussed are few and appear tucked away in paragraph starting on L521 with the study strengths. The ecological design of the study is not the only study limitation. Other examples the authors may want to consider are the potential limitations of using the SVIs, in general, or potential reporting issues of VL that may lead to uncertainty, or limitations of choices in statistical methodology. 

Besides the public policy angle, what insight about VL epidemiology or natural history does the investigation demonstrate? Or what further questions does the investigation raise and what future directions can this study be taken. For example, a future direction could be to conduct spatial regression to further assess the relationship between VL and SVI beyond spatial- or spatio-temporal cluster detection.

**Editorial and Data Presentation Modifications?**

Reviewer #1: O texto foi adequamente escrito e possui pontencial de publicação.

Reviewer #2: The paper needs minor modifications by considering the above comments.

Reviewer #3: Spell out “<“ when used in body of text

L72 “Ecological and time series”, remove the ‘and’

L253 typo

L117 use of semicolon

L102 typo

L321 the syntax in this section needs some tightening and double-check the formatting of all 95%CIs

L348 & L388 spell out “Fig” when in-line text

The sentence starting in L459 seems out of place

Paragraph starting on L473 and L478 can be combined

Table 2: Typo in a heading of a column

Figure 4: suggestion to have one common compass rose and scale bar like in Figure 6

Table 3: If indicated all clusters are statistically significant, then the authors may remove the p-value column.

Supplemental 2 Figure 1: Inclusion of an overall trend line combining all federative units/states would help with comparisons between federative units/states

**Summary and General Comments**

Reviewer #1: (No Response)

Reviewer #2: The paper generally good but the author try to update all the necessary comments particularly in methodology section including variable classifications, definitions and statistical model formulations. Some editorial problem the author must modify as per the format of the journal including tables and figures. Updating recent studies on this area under discussion part and also modify the conclusion.

Reviewer #3: The authors conducted a novel investigation of the relationship between the incidence of a neglected tropical disease and social vulnerability in a region of Brazil with the highest historical prevalence and high degree of social inequalities. The study of health disparities is an important topic and health research priority. The study is thorough and with some improvements it could be considered for publication.

PLOS authors have the option to publish the peer review history of their article (what does this mean?). If published, this will include your full peer review and any attached files.

Reviewer #1: No

Reviewer #2: Yes: Anteneh Asmare Godana (PhD)(BSc in Statistics, MSc in Applied Statistics, Ph.D. in Statistics)

Assistant Professor of Statistics, 

University of Gondar College of Natural and Computational Sciences, 

Department of Statistics 

P.0.Box 196 

Email: antenehstat1988@gmail.com, anteneh.asmare@uog.edu.et

Reviewer #3: Yes: Ian D. Buller, Ph.D., M.A.
---

## [Decision Letter · Decision Letter 1]

16 Oct 2020

Dear Dr. Rodrigues de Moura,

Thank you very much for submitting your manuscript "Space-time risk cluster of visceral leishmaniasis in Brazilian endemic region with high social vulnerability: an ecological time series study" for consideration at PLOS Neglected Tropical Diseases. As with all papers reviewed by the journal, your manuscript was reviewed by members of the editorial board and by several independent reviewers. The reviewers appreciated the attention to an important topic. Based on the reviews, we are likely to accept this manuscript for publication, providing that you modify the manuscript according to the review recommendations. 

Thank you very much for resubmitting your manuscript. It is much improved and only requires a few minor changes. Please take into account the comments of Reviewer 3, Ian D. Buller, Ph.D., M.A., at this stage and resubmit.

Sincerely,

Genevieve Milon

Deputy Editor

Genevieve Milon

Deputy Editor

Thank you very much for resubmitting your manuscript. It is much improved and only requires a few minor changes. Please take into account the comments of Reviewer 3, Ian D. Buller, Ph.D., M.A., at this stage and resubmit.

Reviewer's Responses to Questions

**Key Review Criteria Required for Acceptance?**

**Methods**

-Are the objectives of the study clearly articulated with a clear testable hypothesis stated?

-Is the study design appropriate to address the stated objectives?

-Is the population clearly described and appropriate for the hypothesis being tested?

-Is the sample size sufficient to ensure adequate power to address the hypothesis being tested?

-Were correct statistical analysis used to support conclusions?

-Are there concerns about ethical or regulatory requirements being met?

Reviewer #3: Yes, sufficient revisions after previous review.

**Results**

-Does the analysis presented match the analysis plan?

-Are the results clearly and completely presented?

-Are the figures (Tables, Images) of sufficient quality for clarity?

Reviewer #3: Yes, sufficient revisions after previous review.

**Conclusions**

-Are the conclusions supported by the data presented?

-Are the limitations of analysis clearly described?

-Do the authors discuss how these data can be helpful to advance our understanding of the topic under study?

-Is public health relevance addressed?

Reviewer #3: L522: Great addition about data limitations. If there is evidence of underreporting at the health systems in Northeastern Brazil, then its citation would enhance your limitation section. If there is no evidence or study about health system underreporting then I would recommend adding “potential” before “underreporting of cases” and “missingness” replacing “the missing.” You could combine with the following sentence by removing “Maybe this situation occurs due the” with “possibly due to the.”

**Editorial and Data Presentation Modifications?**

Reviewer #3: “global Moran’s index I” = “global Moran’s I index”

L436: Typo “trough” = “through”’

Table 2: “Miss data = “Missing data”

L369, L485: use of “whose” seems awkward here. Try “and” or removing

L456: Sentence “Similarly, in Brazil” is awkward, try “has been reported” at the end of the sentence. 

L465: Sentence “In Rio Grande do Norte…” is awkward, try removing “it was found” for “there is” 

L487: This is the first mention of Lutzomyia longipalpis so provide the full genus

L515: Suggestion to change “it is urgent to develop researches” to “there is a need to conduct investigations”

S2 Appendix 1: Typo in column 2 row 3 “inclued” = “included” (it appears correctly in track-changed document)

S2 Appendix 3: Typo “ie” = “i.e.,”

**Summary and General Comments**

Reviewer #3: The authors returned with an improved manuscript after considering all suggestions from a previous review. The study remains in the scope of the journal and is an important analysis health disparity. I thank the authors for their collegial and thorough responses. I have few substantive edits for consideration, primarily minor grammatical modification for clarity.

PLOS authors have the option to publish the peer review history of their article (what does this mean?). If published, this will include your full peer review and any attached files.

Reviewer #3: Yes: Ian D. Buller, Ph.D., M.A.
---

## [Editor Report · Decision Letter 2]

24 Nov 2020

Dear Dr. Rodrigues de Moura,

We are pleased to inform you that your manuscript 'Space-time risk cluster of visceral leishmaniasis in Brazilian endemic region with high social vulnerability: an ecological time series study' has been provisionally accepted for publication in PLOS Neglected Tropical Diseases.

Best regards,

Sitara SR Ajjampur

Guest Editor

Genevieve Milon

Deputy Editor

---

## [Editor Report · Acceptance letter]

14 Jan 2021

Dear Dr. Rodrigues de Moura,

We are delighted to inform you that your manuscript, "Space-time risk cluster of visceral leishmaniasis in Brazilian endemic region with high social vulnerability: an ecological time series study," has been formally accepted for publication in PLOS Neglected Tropical Diseases.

Best regards,

Shaden Kamhawi

co-Editor-in-Chief

Paul Brindley

co-Editor-in-Chief
